# Beyond IID: data-driven decision-making in heterogeneous environments

**Omar Besbes**[*] **Will Ma** **Omar Mouchtaki**
Decision, Risk, and Operations Division
Columbia University
{ob2105,wm2428,om2316}@gsb.columbia.edu

## Abstract

In this work, we study data-driven decision-making and depart from the classical identically and independently distributed (i.i.d.) assumption. We present a new framework in which historical samples are generated from unknown and different distributions, which we dub *heterogeneous environments*. These distributions are assumed to lie in a heterogeneity ball with known radius and centered around the (also) unknown future (out-of-sample) distribution on which the performance of a decision will be evaluated. We quantify the asymptotic worst-case regret that is achievable by central data-driven policies such as Sample Average Approximation, but also by rate-optimal ones, as a function of the radius of the heterogeneity ball. Our work shows that the type of achievable performance varies considerably across different combinations of problem classes and notions of heterogeneity. We demonstrate the versatility of our framework by comparing achievable guarantees for the heterogeneous version of widely studied data-driven problems such as pricing, ski-rental, and newsvendor. En route, we establish a new connection between data-driven decision-making and distributionally robust optimization.

## 1 Introduction

In optimization under uncertainty, the desirability of a decision (e.g., inventory) depends on an unknown future outcome (e.g., demand). Typically, past data is collected to be indicative of the future, and hence inform our decision. However, ideal data-driven decision-making requires postulating beliefs about the reliability of the past data, and importantly, whether the future may *deviate* from it. In practice, past data may depend on contexts, some of which can be controlled for, and some of which cannot (unobserved confounders). This may introduce data heterogeneity that is not "correctable."

In this paper, we consider a framework for modeling how the future may deviate from past data. At a high level, it accomplishes three goals: $i$.) capture different forms of future deviation, including no deviation, under the same umbrella; $ii$.) understand the performance of a central policy, which makes the decision that optimizes the *average* objective value over past data, under different forms of unexpected deviation; $iii$.) if the central policy performs poorly, then derive modifications that are robust to different forms of anticipated deviation, and quantify the achievable performance.

This framework is general, capturing different problems, and our analysis leads to insightful problem-specific conclusions. The only assumption is that the policy defined in $ii$.) above can be computed, which requires perfect counterfactual evaluation of any decision on all data points, as opposed to settings where past data are affected by previous actions [5, 16, 33, 41, 42, 43]. The policy described in $ii$.) is widely studied across different fields, known as *Sample Average Approximation (SAA)* [37, 36, 39], although we emphasize that we also derive *new policies beyond SAA* when it falters.

---

[*]Authors ordered alphabetically.

36th Conference on Neural Information Processing Systems (NeurIPS 2022).

## 1.1 Framework description

**Framework (Section 2.1).** Our first contribution lies in studying a framework which models the future outcome as being drawn from an unknown distribution, and past data as being drawn independently from (also unknown) "nearby" distributions. In this way, past data is indicative of the future, and we emphasize that the samples of data can be drawn from different nearby distributions. In accord, we call this setting a *heterogeneous environment*. Using varying definitions of "nearby distribution" (e.g. based on Kolmogorov/Wasserstein distances; see Section 1.3), we analyze different forms of deviation possible between the past and future, and let a radius $\epsilon$ bound the allowed deviation. When $\epsilon = 0$, our framework captures the classical independent and identically distributed (i.i.d.) setting, in which past data are drawn from the same distribution as the future outcome [37, 45, 4]. This framework presents another possible parametrization to interpolate between the i.i.d. setting and the adversarial one and shares conceptually similar goals to previous approaches considering algorithm analysis beyond worst-case [51, 7, 53, 25, 10, 24].

**Performance measure (Section 2.2).** We consider the performance of different data-driven policies, which map past data into a decision for a given problem; SAA defines a feasible data-driven policy for any problem. Regret is measured as the difference in objective between the policy's decision and the optimal decision knowing the future distribution, taking an expectation over the draws of past data (which affect the policy's decision), any intrinsic randomness in the policy, and the outcome realization (which affects the objective of both the policy's decision and the optimal decision). We then take a worst case regret over all possible distributions that could have been chosen for the future outcome and the data points, to evaluate the performance of a fixed data-driven policy. We note that this performance depends on the problem (see Section 1.3 for examples), the number of data points $n$, the definition of "nearby distribution," and the radius $\epsilon$.

## 1.2 General results

Our second contribution lies in developing a set of general results for this framework. A central result we present in Section 3 is a general reduction we develop in Theorem 1 in which we establish the following: asymptotically, as the number of samples goes to $\infty$, the worst-case regret of a policy (in a broad subclass we call sample-size-agnostic) is upper-bounded by that of an alternative problem with two major simplifications: $i.$) the worst-case is now taken over only a single distribution for historical environments as opposed to a sequence of heterogeneous distributions; and $ii.$) the decision-maker (DM) has access to the exact historical distribution. This upper bound can be seen as a uniform distributionally robust optimization problem under a regret performance metric, and offers a systematic way to derive upper bounds for policies of interest. This result is closely related to equation (7) in [46] and we discuss in detail the connections to their result after stating Theorem 1.

We develop problem-independent results that hold whenever the objective function is Lipschitz as a function of the future outcome that materializes (Section 4). In that case, we establish in Theorem 2 that SAA scales linearly in $\epsilon$ for both the Kolmogorov and Wasserstein forms of deviation. We also show that for this class of problems, the Lipschitz constant characterizes the hardness of the instance.

## 1.3 Problem-specific, deviation-specific results

Our third contribution lies in deriving problem-specific conclusions in our general heterogeneous environments. In particular, we consider three classical problems in the context of our framework—newsvendor, pricing, and ski-rental—that have received significant attention in the literature across Operations Research, Economics, and Computer Science in the known-prior, data-driven, distributionally robust, adversarial, and advice-augmented settings. Problem-specific results are obtained by using: $i.$) our general reduction and results to upper-bound their asymptotic worst-case regret, and $ii.$) a combination of algorithm design and the development of impossibility results (lower bounds) on the worst-case regret. We analyze the regret for the SAA policy as well as alternative policies.

**Problem descriptions.** First, we consider the newsvendor problem (Section 4.1), in which a DM must decide how much capacity or inventory to plan in the face of unknown demand. The objective is to minimize the sum of *underage cost*, paid for each unit by which capacity falls short of realized demand, and *overage cost*, paid for each unit of capacity in excess of demand that needs to sit idle. This is a foundational problem in Operations Research, with early focus on the setting where the demand distribution is known [see 47], and since then studied in various distributionally robust

[54, 22, 49] settings but also in data-driven ones [40, 15, 6]. Newsvendor captures the fundamental tradeoff in guessing an unknown value when there are asymmetric consequences for going under/over.

Second, we consider the single-item pricing problem (Section 5.1), in which a DM must find the best price to offer to a customer with unknown willingness to pay (wtp). The objective is to maximize revenue. This is a foundational problem in Economics, with early focus on the setting with known prior [see 48, 52], and more recently in data-driven [21, 29, 2, 1, 17] and distributionally robust [12, 23] settings. Pricing captures the fundamental tradeoff between margin and volume.

Finally, we consider the ski-rental problem (Section 5.2), in which a DM must decide each day whether to buy skis or keep renting, without knowing the length of their trip. The objective is to minimize total cost, where buying eliminates the rental cost going forward. Although typically presented as an online problem that requires a decision to be made every day, any (randomized) buy-or-rent policy can be described by a (randomized) duration for which to rent skis, before committing to buying (if the trip has not ended by then). This is a foundational problem in Computer Science, critical to the development of competitive analysis [see 11] which considers the *adversarial* setting where nothing is known, and recently also important to the development of *advice-augmented algorithms* which receive a prediction on the length of the ski trip [50, 19]. Ski rental captures the fundamental tradeoff between proceeding with a suboptimal process vs. paying the cost of refactoring.

**Definitions of "nearby distribution."** Our framework generally models past data to be drawn from distributions that lie within an $\epsilon$-distance of the future distribution, for *any* notion of distance between probability distributions. For our problems of interest, we focus on two specific distance metrics: Kolmogorov and Wasserstein (see Section 2.1 for formal definitions). This dichotomy suffices to demonstrate the importance of distance metric in determining performance in our problems of interest, and moreover, represents contrasting inductive philosophies about how the future might deviate from the past: Wasserstein allows *every* data point to be erroneous by $\epsilon$, whereas Kolmogorov allows an $\epsilon$ fraction of data points to be *arbitrarily* erroneous. We note that similar notions of "nearby distributions" were considered in [51] in the online learning setting and in [7, Example 5] for the expert selection problem. Our work differs from these two papers because we consider an offline setting and our benchmark is stronger since we focus on the oracle knowing the ground-truth out-of-sample distribution, as opposed to the "best action in hindsight" benchmark.

**SAA vs. robustified policies.** Finally, performance is demarcated by the policy used. We first provide tight asymptotic performance bounds for the SAA policy, which we note is agnostic to the heterogeneity that might arise. We then analyze the best-possible asymptotic performance bounds that can be achieved by any policy when the notion of distance and the radius $\epsilon$ are known.

In Table 1, we present a high level summary of our results across problem classes. We assume that the support of the unknown environment is in $[0, M]$ for some positive real number $M$ which $M$ can be interpreted as parametrizing the "extent" of uncertainty one faces (e.g., maximal values of demand in newsvendor) We track the dependence on both the heterogeneity radius $\epsilon$, as well as $M$. These results are for the asymptotic setting in which $n \to \infty$.

| | | | Kolmogorov | Wasserstein |
|---|---|---|---|---|
| Lipschitz | Newsvendor (§4.1) | SAA | $\Theta(M\epsilon)$ | $\Theta(\epsilon)$ |
| | | best policy | $\Theta(M\epsilon)$ | $\Theta(\epsilon)$ |
| Beyond Lipschitz | Pricing (§5.1) | SAA | $\Theta(M\epsilon)$ | $\Omega(M)$ |
| | | best policy | $\Theta(M\epsilon)$ | $\Theta(\sqrt{M\epsilon})$ |
| | Ski-rental (§5.2) | SAA | $\Theta(M\epsilon)$ | $\Theta(1)$ |
| | | best policy | $\tilde{\Theta}(\epsilon)^*$ | $\Theta(\sqrt{\epsilon})$ |

Table 1: **SAA and achievable performance for different problem classes and heterogeneity balls.** *(Here $\tilde{\Theta}$ provides rate order up to logarathimic factors.)

For newsvendor problems, through a combination of the general result for Lipschitz-continuous problems and lower bounds, we show that SAA actually achieves the optimal rate for asymptotic worst-case regret both as a function of $\epsilon$ and $M$ and that this rate is linear.

We then turn to problems whose objective is not Lipschitz. For the pricing problem, we establish that while under Kolmogorov distance, SAA is near-optimal and the optimal performance is again

of order $M\epsilon$, the picture is starkly different under the Wasserstein distance. In this case, SAA has arbitrarily poor performance. We then show that a robustification procedure that deflates SAA by an appropriate factor can yield the best achievable asymptotic worst-case regret that shrinks to zero at rate $\sqrt{M\epsilon}$. To our understanding, analyzing these non-SAA policies (which are required for good performance on pricing) deviates from standard analyses used in learning theory (see Remark 2).

A second prototypical problem that is not Lipschitz is the ski-rental problem. While SAA was near-optimal for the Kolmogorov distance for pricing and newsvendor, we establish the surprising fact that SAA, even under the Kolmogorov distance, has arbitrarily poor performance. Indeed, its asymptotic worst-case regret scales as $\Theta(M\epsilon)$, while the optimal policy achieves a performance of $\Theta(\epsilon)$. As a consequence, the performance of SAA can be arbitrarily worse than the optimal one as the support grows to infinity. Meanwhile, for the Wasserstein distance, we show that the performance of SAA is also suboptimal. Indeed, SAA incurs a regret of $\Theta(1)$; but by inflating SAA appropriately, one can construct a policy that achieves order $\sqrt{\epsilon}$ performance, which is the best dependence on $\epsilon$. These examples demonstrate that SAA fails for both heterogeneity types for ski-rental, but also more broadly the intricate interplay between problem class and type of heterogeneity.

**Connection to offline data corruption models.** Different notions of offline data corruptions and robustness have been previously studied in statistics [30, 26], learning theory [55, 28, 35, 34, 55, 38, 18, 8, 20] and, more recently in revenue management [13, 12, 14, 23]. Most approaches consider $\epsilon$-contamination settings [30] in which the adversary can shift a small proportion of data arbitrarily far from the true distribution. They consider either oblivious adversaries who fix a *single* corrupted distribution from which samples are generated or adaptive one who observe samples before corrupting them. [8] establishes that adaptive adversaries yield equivalent performance to oblivious ones in many settings. Closest to us are [12, 23] who study optimal auctions. They assume the distribution is common across all past observations; the former focuses on regret and the the latter on a ratio metric when data is generated from MHR distributions close in Kolmogorov distance. The framework we develop is anchored around a setting that allows for *heterogeneous* "nearby" distributions for each past observation, and is general in that it allows us to unify a variety of problems/metrics and highlight how these affect the levels of achievable performance.

**Connection to online and batch learning in non-stationary environments.** Our work also relate broadly to papers studying learnability in changing environment. [27, 9] adopt a universal learning approach and analyze algorithms which are able to achieve vanishing long-term average loss for general data-generation processes. [44, 3, 46] derive generalization guarantees in a setting where the training set distribution may differ from the test set distribution.

## 2 Problem formulation: data-driven decisions in heterogeneous environments

We consider a general stochastic optimization problem in which a DM wants to optimize an objective in a stochastic environment. Formally, we denote by $\mathcal{X}$ the decision space and by $\Xi$ the space of environment realizations. The objective of the DM is represented by a function $g : \mathcal{X} \times \Xi \to \mathbb{R}$ which maps each pair of decision $x \in \mathcal{X}$ and environment realization $\xi \in \Xi$ to an objective value.

The DM faces a subset of possible probability measures $\mathcal{P} \subset \Delta(\Xi)$, where $\Delta(\Xi)$ denotes the space of probability measures supported on $\Xi$. For every probability measure $\mu \in \mathcal{P}$, the goal of the DM is to optimize the objective defined as

$$\mathcal{G}(\cdot, \mu) : \begin{cases} \mathcal{X} \to \mathbb{R} \\ x \mapsto \mathbb{E}_{\xi \sim \mu}[g(x, \xi)]. \end{cases} \tag{1}$$

We extend the definition of $\mathcal{G}(\cdot, \mu)$ to allow for randomized decisions $\rho \in \Delta(\mathcal{X})$, in which case we define $\mathcal{G}(\rho, \mu) = \mathbb{E}_{x \sim \rho}[\mathcal{G}(x, \mu)]$ to also take an expectation over the randomized decision.

Remark that depending on the nature of the problem, the objective could represent for example a function to maximize (profit) or to minimize (cost). In turn, we formally define for every $\mu \in \mathcal{P}$, the optimal objective $\text{opt}(\mu)$ as the objective value achieved by an *optimal action*, given by an *oracle* operator `ORACLE` that is a mapping from $\Delta(\Xi)$ to $\mathcal{X}$.

## 2.1 Heterogeneous environments

In the absence of distributional knowledge on the underlying environment, a common approach consists in assuming that historical samples are independent and identically drawn from a fixed probability measure $\mu$. We relax the assumption that the samples are identically distributed and introduce a generalization of this framework that we refer to as $\epsilon$-*heterogeneity*.

Let $d : \mathcal{P} \times \mathcal{P} \to \mathbb{R}$ be a metric on $\mathcal{P}$. Given a target probability measure $\mu \in \mathcal{P}$ and a level of heterogeneity $\epsilon$, we define the heterogeneity ball anchored around $\mu$ and with radius $\epsilon$ as

$$\mathcal{U}_\epsilon(\mu) := \{\nu \in \mathcal{P}, \text{ s.t } d(\mu, \nu) \leq \epsilon\}.$$

The heterogeneity ball $\mathcal{U}_\epsilon(\mu)$ includes all distributions that are within a distance of $\epsilon$ from $\mu$. Next, we formally define a data-driven decision-making problem in a heterogeneous environment.

**Definition 1** (Data-driven decision-making problem in heterogeneous environment). *A data-driven decision-making problem in a heterogeneous environment is defined by the tuple:* $(\mathcal{X}, \Xi, \mathcal{P}, d, g, \texttt{ORACLE})$, *where $\mathcal{X}$ is the set of possible actions, $\Xi$ is the environment space, $\mathcal{P}$ is the space of measures nature can select from, $d$ is a metric on the space of measures, $g$ is an objective function, and* $\texttt{ORACLE}$ *is a mapping from environment measures to target actions.*

**Focal metrics.** The definition above is general, and the framework can be applied to a variety of metrics. In this work, we aim at developing an understanding of the effect of heterogeneity under various metrics and problem classes. We focus on two prototypical and central distances: *Kolmogorov* and *Wasserstein*. We formally define them in one dimension as follows.

**Definition 2** (Kolmogorov and Wasserstein distances). *Given an environment space $\Xi \subset \mathbb{R}$ and a subset of probability measures $\mathcal{P}$ supported on $\Xi$, the Kolmogorov (resp. Wasserstein) distance $d_K$ (resp. $d_W$) is defined for every $\mu_1, \mu_2 \in \mathcal{P}$ as*

$$d_K(\mu_1, \mu_2) = \sup_{\xi \in \Xi} |F_1(\xi) - F_2(\xi)| \qquad and \qquad d_W(\mu_1, \mu_2) = \int_{\xi \in \Xi} |F_1(\xi) - F_2(\xi)| \, d\xi,$$

*where $F_1$ (resp. $F_2$) is the cumulative distribution associated with $\mu_1$ (resp. $\mu_2$).*

Note that when $\Xi$ is a compact interval of $\mathbb{R}$, both distances are related as for every $\mu_1, \mu_2 \in \mathcal{P}$,

$$d_W(\mu_1, \mu_2) \leq \operatorname{diam} \Xi \cdot d_K(\mu_1, \mu_2). \tag{2}$$

## 2.2 Data-driven policies and performance

**Data-driven policies.** In most settings, the probability measure $\mu$ is unknown to the DM and the optimal objective $\operatorname{opt}(\mu)$ is not achievable. In turn, the DM observes a sequence of historical samples $\xi_1, \xi_2, \ldots, \xi_n$ representing the previous environment realizations. A data-driven policy is then a mapping from the past samples to a (randomized) decision. Define the empirical measure $\hat{\mu}_{\boldsymbol{\xi^n}}$ as

$$\hat{\mu}_{\boldsymbol{\xi^n}}(\xi) := \frac{1}{n} \sum_{i=1}^{n} \mathbb{1}\{\xi = \xi_i\}, \qquad \text{for every } \xi \in \Xi.$$

In a setting in which the order of the samples does not matter (as the one we will consider), to represent a policy taking as input $\boldsymbol{\xi^n} := (\xi_1, \ldots, \xi_n)$, one only needs to consider mappings that take as an input the number of samples and the empirical measure. We define a data-driven policy $\pi$ as a mapping from $\mathbb{N}^* \times \mathcal{P}$ to $\Delta(\mathcal{X})$, the space of measures on the action space $\mathcal{X}$, that associates $(n, \hat{\mu})$ to a distribution of actions $\pi(n, \hat{\mu})$. We let $\Pi$ denote the set of all such mappings. We assume that such a mapping can be made with knowledge of the metric $d$ and radius $\epsilon$. Moreover, we say that $\pi$ is *sample-size-agnostic* if its decision does not depend on $n$, in which case we write $\pi(\hat{\mu})$ for brevity.

A widely-studied policy that achieves excellent performance across a vast range of data-driven decision-making problems is Sample Average Approximation (SAA). Given an empirical distribution $\hat{\mu}$, SAA (denoted by $\pi^{\text{SAA}}$) selects the target action $\texttt{ORACLE}(\hat{\mu})$, the best action associated with $\hat{\mu}$.

**Performance through worst-case regret.** Given an out-of-sample distribution $\mu \in \mathcal{P}$ the best achievable objective is $\operatorname{opt}(\mu)$. For a sequence of historical distributions $\nu_1, \ldots, \nu_n$ in $\mathcal{U}_\epsilon(\mu)$, the expected objective of a policy $\pi$ is

$$\mathbb{E}_{\xi_i \sim \nu_i} \left[ \mathcal{G}\left(\pi(n, \hat{\mu}_{\boldsymbol{\xi^n}}), \mu\right) \right].$$

In turn, we define the expected (absolute) regret of a policy $\pi \in \Pi$ against an out-of-sample distribution $\mu \in \mathcal{P}$ and a sequence of $n$ historical distributions $\nu_1, \ldots, \nu_n \in \mathcal{U}_\epsilon(\mu)$ as

$$\mathcal{R}_n(\pi, \mu, \nu_1, \ldots, \nu_n) := |\mathbb{E}_{\xi_i \sim \nu_i}[\mathcal{G}(\pi(n, \hat{\mu}_{\boldsymbol{\xi}^n}), \mu)] - \mathrm{opt}(\mu)|.$$

We note that we use the absolute value in this definition, as our framework applies to minimization and maximization problem. Once the type of problem is specified (i.e., the operator associated with `ORACLE`), this definition of regret coincides with the usual regret for minimization or maximization problems, as we will see in Sections 4 and 5, when describing specific applications.

In this work, we will assess the performance of a policy based on its *worst-case expected regret*. Given an instance $\mathcal{I} = (\mathcal{X}, \Xi, \mathcal{P}, d, g, \texttt{ORACLE})$, we define the worst-case expected regret of a data-driven policy $\pi \in \Pi$ for a degree of heterogeneity $\epsilon \geq 0$ as

$$\mathfrak{R}^\pi_{\mathcal{I},n}(\epsilon) := \sup_{\mu \in \mathcal{P}} \sup_{\nu_1, \ldots, \nu_n \in \mathcal{U}_\epsilon(\mu)} \mathcal{R}_n(\pi, \mu, \nu_1, \ldots, \nu_n). \tag{3}$$

We finally define the *worst-case asymptotic regret* of a data-driven policy $\pi \in \Pi$ in a $\epsilon$-heterogeneous environment as the number of samples $n$ grows large as

$$\mathfrak{R}^\pi_{\mathcal{I},\infty}(\epsilon) := \limsup_{n \to \infty} \mathfrak{R}^\pi_{\mathcal{I},n}(\epsilon). \tag{4}$$

We sometimes write $\mathfrak{R}^\pi_{K,\infty}(\epsilon)$ or $\mathfrak{R}^\pi_{W,\infty}(\epsilon)$ instead of $\mathfrak{R}^\pi_{\mathcal{I},\infty}(\epsilon)$ when the problem class is specified to highlight the dependence on the Kolmogorov or Wasserstein metrics.

## 3 Reduction to distributionally robust optimization in the asymptotic regime

A significant challenge in the analysis of problem (4) stems from the difficulty to characterize a potentially complex worst-case sequence of historical probability measures $\nu_1, \ldots, \nu_n$ and out-of-sample measure $\mu$. We illustrate in the appendix that analyzing every element in the sequence of historical probability measures as a function of $n$ is a priori challenging as nature will attempt to use different distributions for each of the past environments. We will show that, asymptotically, it suffices to consider cases in which nature selects a common distribution for all of the past environments. Formally, we define for a sample-size-agnostic policy $\pi \in \Pi$ the following optimization problem:

$$\sup_{\mu \in \mathcal{P}} \sup_{\nu \in \mathcal{U}_\epsilon(\mu)} |\mathrm{opt}(\mu) - \mathcal{G}(\pi(\nu), \mu)|. \tag{5}$$

While Problem (5) resembles Problem (3), it differs on two crucial dimensions: $i$.) in Problem (5), nature selects the *same* distribution $\nu$ in the heterogeneity ball for all past environments; $ii$.) in Problem (5), the policy $\pi$ is assumed to know the actual distribution of the past environments $\nu$, whereas in Problem (3), the policy only observed the empirical distribution of past realizations.

Next, we will establish that Problems (5) and (3) are tightly connected under some mild assumptions. We introduce the following definition on the distance $d$ defining the heterogeneity.

**Definition 3** (Empirical triangular convergence)**.** *We say that a distance $d$ on the space of probability measures satisfies the empirical triangular convergence (ETC) property if and only if for every triangular array sequence of probability measures $(\mu_{i,n})_{1 \leq i \leq n, n \in \mathbb{N}^*}$ all belonging to $\mathcal{P}$, we have*

$$\lim_{n \to \infty} d(\hat{\mu}_n, \bar{\mu}_n) = 0 \qquad a.s.,$$

*where $\hat{\mu}_n(\xi) := \frac{1}{n} \sum_{i=1}^n \mathbb{1}\{\xi = \xi_{i,n}\}$ for samples $\xi_{i,n} \sim \mu_{i,n}$ and $\bar{\mu}_n = \frac{1}{n} \sum_{i=1}^n \mu_{i,n}$.*

The ETC property requires that the sequence of empirical distributions converges to the average of ground truth distributions for arbitrary triangular arrays. In what follows we also impose convexity on the metric $d$. We note that both properties are satisfied by many common distances on probability spaces, including the Kolmogorov and Wasserstein distances on compact sets, as we discuss in the appendix. We now state our first main result which establishes a fundamental reduction.

**Theorem 1** (Upper bound reduction)**.** *Given an instance $\mathcal{I} = (\mathcal{X}, \Xi, \mathcal{P}, d, g, \texttt{ORACLE})$, let $\pi \in \Pi$ be a sample-size-agnostic policy. Assume that $g$ is bounded on $\mathcal{X} \times \Xi$ and that $d$ is a convex metric which satisfies the empirical triangular convergence property. Then*

$$\mathfrak{R}^\pi_{\mathcal{I},\infty}(\epsilon) \leq \lim_{\eta \to \epsilon^+} \sup_{\mu \in \mathcal{P}} \sup_{\nu \in \mathcal{U}_\eta(\mu)} |opt(\mu) - \mathcal{G}(\pi(\nu), \mu)|. \tag{6}$$

Theorem 1 establishes, that under some mild conditions, in the asymptotic regime, one may restrict attention to problem (5). This reduction significantly simplifies the analysis of sample-size-agnostic policies[2] as problem (5) does not involve any form of randomness in the input received by the policy, and only allows the adversary to select a single distribution.

**Remark 1** (Relation to [46]). *We note that [46, eq. (7)] presents a finite-time bound on the regret of SAA in a similar setting to ours. Therefore by taking the limit in their bound, one may derive an asymptotic performance guarantee for SAA. In our notation, their result imply that,*

$$\mathfrak{R}_{\mathcal{I},\infty}^{\pi^{SAA}} \leq 2 \cdot \sup_{\mu\in\mathcal{P}} \sup_{\nu\in\mathcal{U}_\epsilon(\mu)} \sup_{x\in\mathcal{X}} |\mathcal{G}(x,\mu) - \mathcal{G}(x,\nu)|. \tag{7}$$

*Theorem 1 differs from (7) because our reduction applies to any policy in the wide range of sample-size-agnostic policies. In particular, this enables us to characterize the minimax optimal asymptotic regret even when SAA is not (rate) optimal (see Sections 5.1 and 5.2).*

We also show in the appendix that Theorem 1 holds whenever $\Xi$ is finite without restricting the policy.

# 4 Lipschitz-continuous problems

In this section, we study a broad class of problems for which the underlying objective function $g(x,\cdot)$ is Lipschitz-continuous as a function of the environment.

**Theorem 2** (Upper bounds for Lipschitz-continuous objectives). *Suppose that $\Xi = [0,M]$ for some $M > 0$. Let $\mathcal{I}_K = (\mathcal{X},\Xi,\mathcal{P},d_K,g,\texttt{ORACLE})$ and $\mathcal{I}_W = (\mathcal{X},\Xi,\mathcal{P},d_W,g,\texttt{ORACLE})$ be instances of the data-driven decision problem under the Kolmogorov and Wasserstein distances, respectively. Assume that for every $x \in \mathcal{X}$, the function $g(x,\cdot)$ is L-Lipschitz-continuous and that $\texttt{ORACLE}$ is either a minimization or maximization oracle. Then for every $\epsilon \geq 0$,*

$$\mathfrak{R}_{\mathcal{I}_K,\infty}^{\pi^{SAA}}(\epsilon) \leq 2L \cdot M \cdot \epsilon \qquad and \qquad \mathfrak{R}_{\mathcal{I}_W,\infty}^{\pi^{SAA}}(\epsilon) \leq 2L \cdot \epsilon.$$

Theorem 2 implies that for this class of problems, the asymptotic regret of SAA vanishes as the radius of the heterogeneity ball goes to $0$. Furthermore, this theorem shows that the dependence of the asymptotic worst-case regret in the heterogeneity parameter $\epsilon$ is at most linear. We show next that this dependence is the best possible for this class of problems and this notion of heterogeneity.

## 4.1 Newsvendor problem

Recall the newsvendor problem described in Section 1.3. Using $M > 0$ to denote an upper bound on demand, we let $\Xi = \mathcal{X} = [0,M]$, and our regret bounds may (or may not) depend on $M$. The objective function is the newsvendor cost which is parameterized by two quantities: $c_u$ the underage cost and $c_o$ the overage cost. The cost of an inventory level $x \in \mathcal{X}$ when observing demand $\xi \in \Xi$ is,

$$g(x,\xi) = c_u (\xi - x)^+ + c_o (x - \xi)^+,$$

where $(\cdot)^+ := \max\{\cdot, 0\}$ is the positive part operator. The measure space $\mathcal{P}$ is the set of probability measures on $\Xi$ without restrictions and for every $\mu \in \mathcal{P}$, we define the cost of the oracle as $\mathrm{opt}(\mu) = \min_{x\in\mathcal{X}} \mathcal{G}(x,\mu)$. Therefore the oracle mapping is defined as $\texttt{ORACLE} : \mu \mapsto \arg\min_{x\in\mathcal{X}} \mathcal{G}(x,\mu)$.

We note that the objective function $g$ satisfies the $L$-Lipschitz-continuity property on the environment realization variable $\xi$ with $L = \max(c_u, c_o)$. Theorem 2 thus directly yields an upper bound on the achievable regret for this class of problems, for both Kolmogorov and Wasserstein, through the regret of SAA. We next show that these dependencies are the best possible, under any policy.

**Proposition 1** (Heterogeneous newsvendor). *For the data-driven newsvendor problem in a heterogenous environment under the Kolmogorov and Wasserstein distances, given any $\epsilon \geq 0$,*

$$\frac{c_u + c_o}{2} \cdot \epsilon \cdot M \quad \leq \inf_{\pi\in\Pi} \mathfrak{R}_{K,\infty}^{\pi}(\epsilon) \leq \mathfrak{R}_{K,\infty}^{\pi^{SAA}}(\epsilon) \leq \quad 2\max(c_u,c_o) \cdot \epsilon \cdot M,$$

$$\frac{c_u + c_o}{2} \cdot \epsilon \quad \leq \inf_{\pi\in\Pi} \mathfrak{R}_{W,\infty}^{\pi}(\epsilon) \leq \mathfrak{R}_{W,\infty}^{\pi^{SAA}}(\epsilon) \leq \quad 2\max(c_u,c_o) \cdot \epsilon.$$

---

[2]Note that considering sample-size-agnostic policies is not very restrictive. As we will see in Section 4 and 5 this class is sufficient to achieve optimal rates for the asymptotic worst-case regret as a function of $\epsilon$.

Proposition 1 first shows that SAA is robust to deviation under multiple forms of heterogeneity and that the regret scales linearly with the radius of the heterogeneity ball for both distances. We show furthermore that the linear dependence is, for the newsvendor problem, the best rate achievable for any data-driven policy. We also note that the lower bound obtained for the newsvendor depends linearly on the Lipschitz constant $L$ through the factor $(c_u + c_o)$.

Beyond newsvendor, the asymptotic worst-case regret of SAA over the general class of $L$-Lipschitz-continuous functions is of the order $\Theta(L \cdot \epsilon)$. This suggests that the performance of SAA may deteriorate for problems in which the objective function $g$ is not smooth.

# 5 Beyond Lipschitz-continuous problems

## 5.1 Pricing problem

Recall the pricing problem presented in Section 1.3. The environment space $\Xi = [0, M]$ represents the set of possible wtp. Similarly, the set of possible prices $\mathcal{X}$ is $[0, M]$. The set of considered probability measures $\mathcal{P}$ is the whole set of probability measures on $[0, M]$ without restriction. The revenue generated by the DM with a price decision $x \in \mathcal{X}$ when facing a customer with wtp $\xi \in \Xi$ is,

$$g(x, \xi) = x \cdot \mathbb{1}\{x \leq \xi\}.$$

Therefore for any probability measure $\mu \in \mathcal{P}$ associated with the cumulative distribution $F$, we have that $\mathcal{G}(x, \mu) = x \cdot (1 - F(x))$. The goal of the DM is to set a price $x$ in order to maximize its revenue. Equivalently, the goal is to minimize the regret against the oracle which posts the optimal price given a wtp distribution. Formally, the oracle operator is defined as, $\texttt{ORACLE} : \mu \mapsto \arg\max_{x \in \mathcal{X}} \mathcal{G}(x, \mu)$.

**Kolmogorov pricing.** Remark that the pricing objective is not Lipschitz-continuous and that one cannot use the argument presented in Section 4.1. We show in the appendix that despite the lack of smoothness of the objective, SAA is still near-optimal in the asymptotic regime under the Kolmogorov distance. We derive an upper bound on the asymptotic worst-case regret of SAA and a matching universal lower bound showing that the best achievable performance is order $\Theta(M\epsilon)$.

**Failure of SAA in pricing with Wasserstein heterogeneity.** In opposition to the strong performance achieved by SAA for pricing against Kolmogorov heterogeneity, our next proposition shows that this result does not hold if one considers instead the Wasserstein heterogeneity ball.

**Proposition 2** (SAA and Wasserstein pricing). *For the data-driven pricing problem in a heterogenous environment under the Wasserstein distance, we have for $\epsilon > 0$ that*

$$\mathfrak{R}_{W,\infty}^{\pi^{SAA}}(\epsilon) = M.$$

Proposition 2 shows that, in stark contrast with Lipschitz problems, for the pricing problem, the asymptotic regret of SAA under Wasserstein heterogeneity does not shrink to zero as $\epsilon$ goes to zero. As a matter of fact, the regret of $M$ is the worst possible, as the revenue of the oracle is bounded from above by $M$ and the revenue of any data-driven policy is bounded from below by $0$. Quite notably, this result holds for any level of heterogeneity $\epsilon$, and an arbitrarily small deviation from the i.i.d. case leads to extremely poor performance, even with infinite data. We next propose a policy which "robustifies" SAA and we analyze it under the Wasserstein distance.

**Robustification of SAA for Wasserstein pricing.** The key challenge in pricing is that for two distributions $\mu$ and $\nu$ such that $d_W(\mu, \nu)$ is arbitrarily small, a price $x$ could have a high revenue for $\mu$ but a low revenue for $\nu$, and the difference in revenue between distributions may be arbitrarily large. This enabled us to construct an instance for which the optimal price for the in-sample distribution $\nu$ generates a low revenue for the out-of-sample distribution $\mu$.

**Remark 2** (Proof Sketch, Comparison with Standard Techniques). *To the best of our understanding, our analysis of policies beyond SAA requires a new technique. Assume $M = 1$ for brevity, and consider the problem of pricing. We need to upper-bound the loss $\mathcal{G}(\texttt{ORACLE}(\mu), \mu) - \mathcal{G}(\pi(\nu), \mu)$. A standard analysis of SAA, which satisfies $\pi^{SAA}(\nu) = \texttt{ORACLE}(\nu)$, would write,*

$$\mathcal{G}(\texttt{ORACLE}(\mu), \mu) - \mathcal{G}(\texttt{ORACLE}(\nu), \mu) \leq 2 \sup_{x \in \mathcal{X}} |\mathcal{G}(x, \mu) - \mathcal{G}(x, \nu)|.$$

*However, for certain decisions $x$, the error $|\mathcal{G}(x, \mu) - \mathcal{G}(x, \nu)|$ can be non-vanishing even if the distance $d_W(\mu, \nu)$ is vanishing. Our fix is the following. For a generic policy $\pi$, we write*

$$\mathcal{G}(\text{ORACLE}(\mu), \mu) - \mathcal{G}(\pi(\nu), \mu) \leq (\mathcal{G}(\text{ORACLE}(\mu), \mu) - \mathcal{G}(\pi(\mu), \nu))$$
$$+ (\mathcal{G}(\text{ORACLE}(\nu), \nu) - \mathcal{G}(\pi(\nu), \mu)) \quad (8)$$

*The key observation is that if we set $\pi(\nu)$ not to $\text{ORACLE}(\nu)$, but to $\text{ORACLE}(\nu) - \delta$, then both large parentheses in (8) can be (identically) upper-bounded using the relation we introduce in Proposition 3.*

**Proposition 3.** *Let $\mu$ and $\nu \in \mathcal{P}$ and let $x_1$ and $x_2$ be two prices such that $x_1 < x_2$. We have that,*

$$\mathcal{G}(x_2, \nu) - \mathcal{G}(x_1, \mu) \leq (x_2 - x_1) + M \cdot \frac{d_W(\mu, \nu)}{x_2 - x_1}.$$

Proposition 3 unveils an interesting tradeoff faced by a DM who prices in a heterogeneous environment. By selecting appropriately the difference between two prices $x_1$ and $x_2$, one can ensure that the gap in revenues for two different distributions can be controlled by the Wasserstein distance.

We next restrict attention to the class of policies which additively *deviate* from the action selected by SAA and leverage Proposition 3 to define the correct deviation parameter.

**Definition 4** ($\delta$-SAA policies). *For any one-dimensional problem we say that a policy is a $\delta$-SAA policy if for every empirical distribution $\hat{\mu}$, the policy selects the closest action in $\mathcal{X}$ to $\text{ORACLE}(\hat{\mu}) + \delta$, for some $\delta \in \mathbb{R}$ that could be positive or negative. We denote this policy by $\pi^{SAA(\delta)}$.*

We note that when $\delta = 0$, we recover the usual SAA policy. Our next result establishes the best possible regret dependence on $\epsilon$ and $M$ for the class of pricing problems, using a $\delta$-SAA policy.

**Theorem 3** (Deviation for Wasserstein pricing). *For the data-driven pricing problem in a heterogenous environment under the Wasserstein distance, for any $\epsilon > 0$, let $\tilde{\delta} = -\sqrt{M \cdot \epsilon}$. Then,*

$$\frac{1}{4} \cdot \sqrt{M \cdot \epsilon} \leq \inf_{\pi \in \Pi} \mathfrak{R}^\pi_{W,\infty}(\epsilon) \leq \mathfrak{R}^{\pi^{SAA(\tilde{\delta})}}_{W,\infty}(\epsilon) \leq 4 \cdot \sqrt{M \cdot \epsilon}.$$

Theorem 3 shows that by appropriately deflating the price posted by SAA, one is able to be robust against Wasserstein heterogeneity and achieve a worst-case asymptotic regret that vanishes to $0$ as the degree of heterogeneity $\epsilon$ goes to $0$. This proves that one may operate efficiently for pricing under Wasserstein heterogeneity as $\epsilon$ goes to $0$. Our result establishes that in contrast with Lipschitz problems, it is now impossible to achieve a linear dependence in $\epsilon$ (since $\sqrt{\epsilon} > \epsilon$), with the regret growing at a rate of $\Omega(\sqrt{M \cdot \epsilon})$ under any policy. A small radius $\epsilon$ has a significantly higher impact in pricing problems than in, e.g., newsvendor problems.

We note that the deflated policy proposed in Theorem 3 improves over SAA when one can make use of the knowledge of $\epsilon$. It is worth noting that SAA does not need such knowledge. Furthermore, one can show that this deflated policy incurs a $\Omega(\sqrt{\epsilon})$ regret for pricing under Kolmogorov distance and thus performs poorly compared to SAA in that setting. A natural question would be to understand the best achievable performance without knowledge of $\epsilon$ and/or $d$.

## 5.2 Ski-rental problem

We now consider the ski-rental problem in which renting skis costs \$1 per unit of time while buying them costs \$$b$ up-front, for some real value $b$. The environment space $\Xi$ represents the set of possible lengths of the ski trip and a decision $x$ represents the duration after which skis should be bought (if the ski trip has not ended by that time). We call $x$ the *rental duration*. Let $\mathcal{X} = \Xi = [0, M]$, where we note that setting $x = M$ indicates that skis should never be bought. The set of probability measures $\mathcal{P}$ is the whole space of probability measures supported on $[0, M]$. For every rental duration $x \in \mathcal{X}$ and any trip length $\xi \in \Xi$, the cost incurred is

$$g(x, \xi) = \xi \cdot \mathbb{1}\{\xi \leq x\} + (b + x) \cdot \mathbb{1}\{\xi > x\}.$$

Finally, as the goal is to minimize cost we have that, $\text{ORACLE} : \mu \mapsto \arg\min_{x \in \mathcal{X}} \mathcal{G}(x, \mu)$.

**Wasserstein ski-rental.** We proved in Section 5.1 that SAA performs arbitrarily poorly in pricing under Wasserstein heterogeneity. We now show that SAA falters similarly for the Wasserstein ski-rental problem. Furthermore, we design and analyze a $\delta$-SAA policy which inflates the action selected by SAA (i.e., $\delta > 0$) and achieves the best rate possible as a function of the heterogeneity radius. Formally, we show the following.

**Theorem 4.** *For the data-driven ski-rental problem in a heterogenous environment under the Wasserstein distance, we have for $\epsilon > 0$ that,*

$$\mathfrak{R}^{\pi^{SAA}}_{W,\infty}(\epsilon) = \Theta(1) \qquad and \qquad \inf_{\pi \in \Pi} \mathfrak{R}^{\pi}_{W,\infty}(\epsilon) = \Theta\left(\sqrt{\epsilon}\right).$$

Theorem 4 formalizes the failure of SAA for ski-rental under Wasserstein distance. A notable fact about the ski-rental problem under Wasserstein heterogeneity is that both the regret of SAA and of the optimal data-driven decision do not scale with the size of the support. For pricing, SAA scaled linearly with $M$ and the optimal policy scales in $\sqrt{M}$. The fact that $\mathfrak{R}^{\pi^{SAA}}_{W,\infty}(\epsilon) = O(1)$ in ski-rental requires a separate non-trivial proof in Theorem 4.[3]

**Kolmogorov ski-rental.** Under the Kolmogorov distance, we had seen that SAA was near-optimal for both the newsvendor and pricing problems. We establish next that the performance of SAA for the ski-rental problem under Kolmogorov distance is, surprisingly, highly suboptimal.

**Theorem 5.** *For the data-driven ski-rental problem in a heterogenous environment under the Kolmogorov distance, we have for $\epsilon > 0$ that,*

$$\mathfrak{R}^{\pi^{SAA}}_{K,\infty}(\epsilon) = \Theta(M \cdot \epsilon) \quad and \quad \inf_{\pi \in \Pi} \mathfrak{R}^{\pi}_{K,\infty}(\epsilon) = \tilde{\Theta}(\epsilon),$$

*where $\tilde{\Theta}$ provides rates order up to logarithmic factors.*

Theorem 5 shows that the asymptotic worst-case regret of SAA scales linearly with the radius of heterogeneity $\epsilon$ and with the size of the support $M$. We also characterize the rate of the optimal asymptotic worst-case regret as a function of $\epsilon$. The upper bound on the asymptotic regret is obtained in [19] through a variant of the SAA policy, which caps the maximum number of days to rent. We complement this result by providing a matching lower bound. Note that, in contrast to the pricing and newsvendor problems, the scaling of SAA is not optimal in $M$. As the scale of the support grows, the asymptotic worst-case regret of SAA is considerably worse than the optimal achievable rate.

## 6 Conclusion

All in all, the present results offer a systematic way of understanding and quantifying the implications of operating in heterogeneous environments. Our framework enables us to develop a common language to compare achievable performance across central classes of problems and to unveil novel insights about the performance of a central policy, Sample Average Approximation, when slightly deviating from the widely studied i.i.d. regime. In settings where SAA fails, we also design robustification procedures achieving rate-optimal asymptotic guarantees. A key takeaway of this analysis across a broad class of problems and for different heterogeneity structures is that it is necessary to understand the structure of the problem we are facing but also the nature of the heterogeneity when designing data-driven policies that are robust to these environments.

To derive our performance guarantees, we established a crucial connection between data-driven decision making in heterogeneous environments and distributionally robust optimization. While this work leverages this connection essentially to derive bounds on achievable performances, we believe that this connection may be of interest to develop new policies from first principles.

This work also opens many additional questions. First, our analysis characterizes the performance in the asymptotic regime where the sample size grows large, and could be complemented by quantifying the performance of policies with finite samples in heterogeneous environments. Furthermore, our results for Lipschitz problems pin down a key driver of performance, whereas isolating the properties of the policies and the framework elements that drive different levels of achievable performance beyond the Lipschitz case remains open. Another key question would be to understand whether there exists a "best of both worlds" policy that does not use the knowledge of the type of heterogeneity and performs well across heterogeneity types (as opposed to the robustified policy designed for pricing and ski-rental) . Finally, we believe that additional exciting and practical complement to this work include incorporating contexts and deriving statistical tests to characterize the type of heterogeneity along with its radius and provide an empirical validation of the procedures developed here.

---

[3]The fact that $\mathfrak{R}^{\pi^{SAA}}_{W,\infty}(\epsilon) \leq M$ in pricing was vacuous.

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
