# OpenReview forum: "Beyond IID: data-driven decision-making in heterogeneous environments"
_NeurIPS.cc/2022/Conference — NeurIPS 2022 Accept_

### Official Review · Reviewer_Jymp · 2022-07-02

**Rating:** 7
**Confidence:** 4
**Soundness:** 4 excellent
**Presentation:** 4 excellent
**Contribution:** 3 good

**Summary:**

The authors study decision-making where the training data may each have been generated from a different distribution, provided all samples are still independent and the distributions are all close in some metric. For certain convex metrics, they upper bound asymptotic regret by a worst-case, data-independent notion. For Kolmogorov and Wasserstein distances, they apply this to 3 optimization problems and provide matching lower bounds. They also show that empirical risk minimization is suboptimal, and show that an algorithm with oracle knowledge of the constraint on training data can be optimal.

**Questions:**

I have no questions beyond the limitations I raised above.

**Limitations:**

I see no negative societal impacts of this theoretical work.

**Strengths And Weaknesses:**

## Summary of Review

The paper is clearly written and provides compelling results with matching upper and lower bounds for well-studied problems. The framing of the related literature and novelty of the data-generating process is lacking and requires improvements, and I believe the interpretation of the results (and their limitations) could be made more clear. All of my proposed changes are non-technical, and so I believe the authors can successfully make them in the revision period.

## Soundness

In equation 9, triangle inequality (first line) and convexity (third line) are used for the metric $d$, and hence all mentions of the metric through the paper need to be appropriately qualified with these properties; this includes the statement of Theorem 1.

Lines 670--671, I don't think the argument is quite rigorous (the argument for equality (c) is intuitive but not clear that it's precise). Fortunately, it follows from first principles.\
Since the objective is linear in $\beta$:
$$
   \inf_{\pi\in\Pi} \sup_{\mu\in \{\mu_-, \mu_+\}} E_{\mathbf{\zeta} \sim \bar\nu} \mathcal{G}(\pi(n, \hat \mu_{\mathbf{\zeta}}), \mu) - \mathrm{Opt}(\mu) = \inf_{\pi\in\Pi} \sup_{\beta\in \Delta(\mu_-, \mu_+)} E_{\mu\sim\beta} E_{\mathbf{\zeta} \sim \bar\nu} \mathcal{G}(\pi(n, \hat \mu_{\mathbf{\zeta}}), \mu) - \mathrm{Opt}(\mu)
$$
By Fubini (everything is bounded):
$$
  =  \inf_{\pi\in\Pi} \sup_{\beta\in \Delta(\mu_-, \mu_+)} E_{\mathbf{\zeta} \sim \bar\nu} E_{\mu\sim\beta} \mathcal{G}(\pi(n, \hat \mu_{\mathbf{\zeta}}), \mu) - \mathrm{Opt}(\mu)
$$
By Jensen's:
$$
  \geq  \sup_{\beta\in \Delta(\mu_-, \mu_+)} E_{\mathbf{\zeta} \sim \bar\nu} \inf_{\pi\in\Pi} E_{\mu\sim\beta} \mathcal{G}(\pi(n, \hat \mu_{\mathbf{\zeta}}), \mu) - \mathrm{Opt}(\mu).
$$
At this point it is precise to argue that this $\inf_{\pi\in\Pi}$ is achieved at a pointmass for the best $x\in\mathcal{X}$ with respect to $\beta$, since there are no restrictions whatsoever on $\Pi$.

Beyond this, I checked Appendices A-C carefully, and Appendix D at a high level; I found no issues with correctness.

## Novelty

I believe that the authors overstate the novelty of their framework, claiming “Our first contribution lies in introducing a framework which models the future outcome as being drawn from an unknown distribution, and past data as being drawn independently from (also unknown) "nearby" distributions.” At a high level, nearly all of classical statistical robustness is focused on performance within a ball under some metric on probability distributions. Much more specifically, the data-generating process they introduce is not novel. https://arxiv.org/abs/1104.5070 introduce the setting where an adversary gets to select a different data-generating distribution on each round from within a (potentially changing over time) set of allowed distributions. One such choice for this set is obviously the balls considered in the present paper. https://arxiv.org/abs/2007.06552 also study this setting, providing an optimal algorithm with optimal dependence on the time horizon; their Example 5 exactly contains the setting of this work (for any distance on distributions). https://arxiv.org/abs/2102.08446, https://arxiv.org/abs/2202.08549, and https://arxiv.org/abs/2202.04690 also consider this setting for the ball defined by a specific measure of distance based on density ratios, providing optimal algorithms and characterizing optimal performance. Finally, the authors discuss offline corruption but neglect the significant literature on online corrupted data; this connection is less relevant, however, since these results often focus on the data being corrupted rather than the distribution being corrupted.

It is worth noting that the authors motivate their move to the asymptotic regime by saying it is “a significant challenge” to study the minimax game for finite time horizon, yet this is exactly what all of the above work does. The above works also allow for an adaptive adversary, that can select the distribution on each training data point by reacting to the player and the random realizations of data; of course their upper bounds would also apply to an oblivious (and independent) adversary like the one studied in this work. The present work considers a single decision, rather than a sequence of decisions, and so not all results are directly comparable; however, the significant study of “near IID” in the online learning literature warrants comparison from the authors.

## Interpretation of results

The definition of SAA is precisely Oracle($\hat\mu$). Throughout the proofs, however, it is used on arbitrary distributions (of course, an arbitrary distribution could be concentrated on n points and be a sample average). This leads to statements like $\pi^{\mathrm{SAA}}(\nu)$, which is then used as Oracle($\nu$). Clearly, however, this policy is not data-dependent -- it acted on a distribution. This is not an issue with the results (they really do bound SAA acting on $\hat\mu$, asymptotically) or the proofs (everything goes through with a relabelling to Oracle($\nu$)). It is purely an issue with interpretation and understanding of what's going on "under the hood". It seems this is all a consequence of the (unproven, but intuitive) claim that for independent data any optimal algorithm need only see $\hat\mu$ and that for the notion of performance, all that matters is the average asymptotically (this part is proven). However, the closeness of the empirical oracle asymptotically and the asymptotic oracle is less surprising when one writes them down this way. In summary, my suggestion is just be clear everywhere that the algorithm considered is Oracle($\hat\nu$), rather than calling it SAA.

Theorem 3 is stated as though this is a strict improvement over SAA, however it requires advance knowledge of $\epsilon$ (which is not true for SAA). Obviously, for robustness we do not expect to know $\epsilon$, and I think this adaptivity should be discussed (leaving it as an open problem). It's worth noting that much of the earlier literature I mentioned above (missing from this paper) focuses on adapting to $\epsilon$ for other problems.

## Minor comments

- Lemma 3 is so standard it's on Wikipedia, no need to reprove.
- The proof of Prop 7 is very similar to that of Prop 1, with actual word-for-word repetition. Making these both follow from some more general lemma or intermediate result would be preferable.

## Typos / Notation

- I see no reason to let the optimization of g be “arbitrary”; pick max or min as a convention, the other case trivially will also hold by swapping the sign.
- Expectations of the form $\mathbb{E}_{\zeta_i \sim \nu_i}$ (such as right after line 204) are somewhat sloppy notation. More preferable is to define some notation using $\mathbf{\zeta^n}$ and $\mathbf{\nu^n}$ (appropriately defined).
- Line 610, should be $\nu_{i,n} \in \mathcal{U}_{\epsilon}(u_n)$.
- Line 614, should just be $\bar\nu_n = (1/n)\sum_{i=1}^n \nu_{i,n}$.
- Line 652, should quantify over $\epsilon$ before doing algebra.

---

> ### Author Response · Authors · 2022-08-02
> **Response to Reviewer Jymp**
>
> We  are very grateful to  the reviewer for the time spent on our paper, and the very detailed and constructive comments which considerably helped us improve the manuscript.
>
> *In equation 9 ...*
>
> **A:** We thank the reviewer for this comment.
> In the revised version of the paper, we explicitly mention the use of the triangular inequality in equation (9)  and we add the convexity as an assumption in the statement of our theorem. We note that convexity is satisfied by Kolmogorov/Wasserstein distances.
>
> *Lines 670--671, I don't think the argument is quite rigorous ...*
>
> **A:** We thank the reviewer for the careful reading and the constructive feedback. The argument for equality $(c)$ was indeed not precise enough. In the revised version of the paper, we fully develop the argument. (see ln. 758-764)
>
> *I believe that the authors overstate...*
>
> **A:** Thank you for pointing us to these related papers.
> In the revised version, we have carefully reworded several parts to avoid giving this (false) impression that we invented the idea of data-generating distributions that "drift" from the true distribution,
> emphasizing more the analysis that we conduct for these particular problems. We also  relate to these previous studies at various junctions: the papers on smoothed analysis are mentioned in passing when discussing the connection to the beyond worst-case theme when describing our contribution (see l. 44-47),  and the papers by Rakhlin et al. (2011) and Bilodeau et al. (2020), which are closer,  are discussed when describing our definition of ``nearby distribution'' (see l. 114-118).
>
> Nonetheless, we do want to highlight what is unique about our analysis of this framework.  First, our philosophical motivation for it is data that is "heterogeneous" from a true distribution which is unknown and fixed, as opposed to a true distribution that is "drifting" over time. We are also not using this framework to restrict an adversary's choice in worst-case analysis, as in some of the smoothed analysis papers.
>
> *It is worth noting that...*
>
> **A:** We consider the asymptotic regime because, even in the limit, the error is generally not zero, and leads to new insights about the robustness of the SAA policy and when one might need to deviate from it. This rich limiting behaviour is to contrast with the setting studied by  Rakhlin et al. (2011) and Bilodeau et al. (2020), in which a no-regret policy can still be derived and thus the limiting behaviour does not completely discriminate between "good" and "bad" algorithms. We also note that we derive in Theorem 6 a finite sample guarantees in the case where $\Xi$ is finite.
>
> Results derived in previous work cannot be directly compared to ours because the benchmark considered in our work is the oracle with knowledge of the out-of-sample distribution while the papers mentioned above focuses on the ``best action in hindsight''. The closest paper related to our benchmark is one that referee 3 brought to our attention [Mohri and Munoz Medina (2012)](https://arxiv.org/abs/1205.4343), and we now contrast our results with theirs. (See remark 1 ln. 265-272)
>
> *The definition of SAA...*.
>
> **A:**
> Thanks for the suggestion!
> In the revised version, we have added an extra equality in all the mathematical expressions involving $\pi^{SAA}(\nu)$ to replace it by $Oracle(\nu)$, and added an explanation line to remind the reader that both are equivalent.
>
> *Theorem 3 is stated as...*
>
> **A:**  Indeed, our work aims at understanding the asymptotic performance of SAA and at comparing it with the asymptotic performance of the best data-driven algorithm knowing $\epsilon$ and $d$. An interesting follow-up question would be to understand whether one could derive a minimax optimal adaptive policy which does not know $\epsilon$ and $d$. In the revised version, we emphasize these aspects after the statement of Theorem 3 (l 348-351) and discuss more in the conclusion the open questions left by this paper (l. 388-411).
>
> * *Lemma 3...*
>
> **A:** We corrected this and instead cite Williams D. (1991), *Probability with martingales*.
>
> * *The proof of Prop 7...*
>
> **A:** We agree with the reviewer.  We did not fix this in the revision for now, since it requires a change in structuring for some proofs.
>
> * *I see no reason to...*
>
> **A:** Thanks for the comment.  We did think carefully about this and thought the advantages of being able to read pricing as a "maximization" problem and newsvendor/ski rental as "minimization" problems (and only having to deal with positive quantities) outweighted the advantages of having a consistent direction.  In the interest of not changing too much, we decided to keep it this way in the revision.
>
> * *Expectations of the form...*
>
> **A:**
> We apologize for this.  We did not fix it for now since it requires defining some new notation.
>
> * *Line 610 ...*
>
> * *Line 614 ...*
>
> * *Line 652 ...*
>
> **A:** We have corrected these.

---

> > ### Comment · Reviewer_Jymp · 2022-08-07
> > **Happy with author responses**
> >
> > Thanks to the authors for the very clear responses and updates in the revised version, it made it very easy to check that my comments have been addressed. It also seems that the other reviewers' insightful comments were addressed. I have updated my score from 6 to 7 accordingly.

---

### Official Review · Reviewer_mWJA · 2022-07-10

**Rating:** 7
**Confidence:** 3
**Soundness:** 4 excellent
**Presentation:** 4 excellent
**Contribution:** 4 excellent

**Summary:**

In this work, the authors proposed a framework to model learning problems with  heterogeneous environments. The worst-case regret is analyzed in the asymptotic case. Specifically, an upper bound is derived via considering a simpler problem, and several lower bounds are derived for three specific problems.

**Questions:**

(1) Ln.59: "DM" is used before its definition.

(2) Ln.133: "demonstrates" -> "demonstrate".

(3) Ln.205-209: In the definition of the absolute regret, the authors mentioned that the absolute value is used to combine both the maximization/minimization cases. Does this hint that the expected objective value of a policy is always smaller/larger than opt(mu) when we consider the maximization/minimization problem? I wonder if it is possible that v_i are "very good" such that the expected objective value of pi is better than opt(mu)? I understand that this does not affect the worst-case regret in Eq.(3).

(4) In Eq.(6), does the reverse inequality hold if we set eta = epsilon? Also, I am a little confused on the necessity of the limit. I think the right-hand side is decreasing in eta?

**Limitations:**

See my comments in the "questions" section.

**Strengths And Weaknesses:**

The framework in this work is interesting and important in many areas (operations research, economics and management science). I think this work is interesting to audiences in robust optimization, machine learning and operations research fields. The presentation of results is very clear and it is easy to follow. I have not read the appendix due to the time limit, but it seems to me that the theoretical results in this work are correct and sound. The only (minor) weakness may be the lack of non-asymptotic results.

---

> ### Author Response · Authors · 2022-08-02
> **Response to Reviewer mWJA**
>
> We thank the reviewer for the  questions and comments and for the time spent reviewing our work.
>
> *The framework in this work is interesting and important in many areas (operations research, economics and management science). I think this work is interesting to audiences in robust optimization, machine learning and operations research fields. The presentation of results is very clear and it is easy to follow. I have not read the appendix due to the time limit, but it seems to me that the theoretical results in this work are correct and sound. The only (minor) weakness may be the lack of non-asymptotic results.*
>
> **A:** We thank the reviewer for the careful comments. We agree with the reviewer that an interesting follow-up question consists in understanding the rate of convergence of various policies towards the best achievable limiting performance characterized in this work.
>
> 1. *Ln.59: "DM" is used before its definition.*
>
> **A:** We corrected it.
>
> 2. *Ln.133: "demonstrates" -> "demonstrate".*
>
> **A:** We corrected it.
>
> 3. *Ln.205-209: In the definition of the absolute regret, the authors mentioned that the absolute value is used to combine both the maximization/minimization cases. Does this hint that the expected objective value of a policy is always smaller/larger than opt(mu) when we consider the maximization/minimization problem? I wonder if it is possible that $v_i$ are "very good" such that the expected objective value of pi is better than opt(mu)? I understand that this does not affect the worst-case regret in Eq.(3).*
>
> **A:** In examples of maximization or minimization, $opt(\mu)$ is the best out-of-sample expected performance one can hope for. Even if the $\nu_i$'s are ``very good," the important thing to note is that the policy $\pi$ is judged on its performance when facing measure $\mu$. By definition, when facing measure $\mu$, the best performance achievable is $opt(\mu)$.
>
> 4. *Eq.(6), does the reverse inequality hold if we set eta = epsilon? Also, I am a little confused on the necessity of the limit. I think the right-hand side is decreasing in eta?*
>
> **A:** Great question! In general, the reverse inequality will not hold if we do not add additional conditions on the space of policies. Consider for example a simple setting where $\mathcal{X}=\Xi=\{0,1\}$ and the objective is to maximize $g(x,\xi)=x$.
>     Let $\pi$ be the policy that takes the better action 1, *unless* input $\nu$ is precisely the distribution that is 1 w.p.~$1/\sqrt{2}$ and 0 otherwise.
>     Clearly, $R_{\mathcal{I},\infty}^{\pi}(\epsilon)=\limsup_{n\to\infty}R_{\mathcal{I},\infty}^{\pi}(\epsilon)=0$
>  since for any $n$, policy $\pi$ will never take action 0 (because $\sqrt{2}$ is irrational).  Meanwhile, $\sup_{\mu \in \mathcal{P}} \sup_{\nu\in \mathcal{U}_\eta(\mu)} | opt(\mu) - \mathcal{G}(\pi(\nu),\mu) |=1$ even if $\eta=0$, because both $\nu$ and $\mu$ can equal this distribution.

---

> > ### Comment · Reviewer_mWJA · 2022-08-07
> > **Post-rebuttal**
> >
> > I would like to thank the authors for responding to my questions. I have read all reviews and responses, and I will not change my score.

---

### Official Review · Reviewer_4sFh · 2022-07-11

**Rating:** 7
**Confidence:** 2
**Soundness:** 3 good
**Presentation:** 4 excellent
**Contribution:** 3 good

**Summary:**

This paper presents a framework for analyzing data-driven decision-making across heterogeneous environments. The decision-making problems of interest include the newsvendor problem, the single-item pricing problem and the ski-rental problem.

Specifically, when future data distribution and past distributions are nearby in Kolmogorov or Wasserstein distances, the paper derives regret bounds where regret is against optimal policies that have knowledge of the future distribution. To facilitate the analysis, the paper considers an asymptotic regime in which a sequence of past distributions converges to one distribution which is assumed to be known. The paper provides bounds for both general Lipschitz-continuous problems and specific decision-making problems mentioned above.

**Questions:**

Does the robustified SAA policy also work for Kolmogorov pricing?

**Limitations:**

Open questions are addressed as limitations in Section 6. Potential societal impact is not discussed.

**Strengths And Weaknesses:**

Originality: The paper discusses related works on data corruptions and robustness. I am not familiar with the literature on learning across heterogeneous environments or out-of-sample generalization for the specific data-driven decision problems considered in this paper. However, for supervised learning, there appears to be a rich literature on domain adaptation and transfer learning, which seems related to this paper. For example, Mohri and Medina (2012) provide learning bounds in drifting environments; Ben-David, et al. (2010) characterizes transfer learning across different domains. The framework proposed in this paper seems novel.

Quality: The results seem correct to me though I did not check the proofs in the appendix.

Clarity: The paper is well-written and easy to follow. I have some minor comments:

(1) The three contributions in Section 3 are scattered in different subsections. It is a bit confusing when "our third contribution" appears near the top of Section 1.2 (line 70).

(2) The acronym "DM" has not been introduced yet in line 59.

(3) In line 111, at this point it is unclear what the support of the unknown environment and $M$ mean for the specific DM problems.

Significance: This paper provides a framework that unifies a collection of results for out-of-sample learning in different decision-making problems. Again, I am not very familiar with the literature for the decision-making problems of interest, but the results seem interesting and thorough and the paper could be of interest for people studying data-driven decision-making, transfer learning or learning theory. One limitation is that the analysis is done in the simplified asymptotic regime; it would be interesting to see results for a sequence of past distributions, similar to Theorem 1 of (Mohri and Medina, 2012).

- Mohri, Mehryar, and Andres Muñoz Medina. "New analysis and algorithm for learning with drifting distributions." International Conference on Algorithmic Learning Theory. Springer, Berlin, Heidelberg, 2012.
- Ben-David, Shai, et al. "A theory of learning from different domains." Machine learning 79.1 (2010): 151-175.

---

> ### Author Response · Authors · 2022-08-02
> **Response to Reviewer 4sFh**
>
> We thank the reviewer for the questions and for bringing our attention to very relevant references which helped us better  position our work.
>
> *Originality: The paper discusses related works on data corruptions and robustness. I am not familiar with the literature on learning across heterogeneous environments or out-of-sample generalization for the specific data-driven decision problems considered in this paper. However, for supervised learning, there appears to be a rich literature on domain adaptation and transfer learning, which seems related to this paper. For example, Mohri and Medina (2012) provide learning bounds in drifting environments; Ben-David, et al. (2010) characterizes transfer learning across different domains. The framework proposed in this paper seems novel.*
>
> **A:** We thank the referee for bringing these references to our attention. We found the reference  Mohri and Medina (2012) particularly relevant, and now connect to it at various junctions in the paper. In the introduction, we refer to it when discussing Theorem 1 and nearby distributions. We also refer to it after we present our Theorem 1 and contrast some of our results with theirs. (See Remark 1 ln. 265-272)
>
> 1. *The three contributions in Section 3 are scattered in different subsections. It is a bit confusing when "our third contribution" appears near the top of Section 1.2 (line 70).*
>
>  **A:** Thank you for your comment. We have reorganized the text so that each subsection starts off with a high level description of the contribution and then delves in more detail.
>
> 2. *The acronym "DM" has not been introduced yet in line 59.*
>
> **A:** We corrected it.
>
> 3. *In line 111, at this point it is unclear what the support of the unknown environment and  mean for the specific DM problems.*
>
> **A:** Thank you for this comment. We have added a comment about the fact that $M$ can be interpreted as parametrizing the extent of uncertainty one faces.
>
> *Does the robustified SAA policy also work for Kolmogorov pricing?*
>
> **A:** Thank you for this question. The  robustified SAA policy still achieves limiting minimax regret that shrinks to zero as $\epsilon$ converges to zero, but it incurs a regret of $\Omega(\sqrt{\epsilon})$ under Kolmogorov heterogeneity (whereas a non-robustified SAA achieves $\Theta(\epsilon)$, which is better in the case of Kolmogorov uncertainty). A natural follow-up question is whether one could derive a policy that achieves best of both world (that is to say, the best possible rate for Kolmogorov and Wasserstein without knowing the nature of heterogeneity $d$). In the revised version of the paper, we discuss these limitations of the robustified policy (see ln. 370-374). In the conclusion, we formulate as an open question the design of policies achieving good performances across notions of heterogeneity (see ln. 428-431).

---

> > ### Comment · Reviewer_4sFh · 2022-08-07
> > **Thank you for your response**
> >
> > I would like to thank the authors for their response. I have read the response and the other reviews and would like to keep my score the same.

---

### Official Review · Reviewer_xMuo · 2022-07-12

**Rating:** 7
**Confidence:** 3
**Soundness:** 4 excellent
**Presentation:** 4 excellent
**Contribution:** 3 good

**Summary:**

The paper studies data-driven decision-making in heterogeneous environments; in particular, the goal is to solve a stochastic optimization problem based on previous observations that were sampled from a Wasserstein/Kolmogorov ball around the target distribution. Main results analyze the regret of sample average approximation policies. The regret rates vary significantly depending on the notion of heterogeneity and problem class.

**Questions:**

A natural question that arises is whether it is possible to somehow fix SAA to get the optimal rates for the ski-rental problem. I don't think this is necessary though.

**Limitations:**

The authors do not discuss limitations and potential negative societal impact explicitly. I don't see what they could say about negative societal impact, but perhaps they could add some discussion about limitations.

**Strengths And Weaknesses:**

The paper is exceptionally clearly written, well-motivated, and (to my admittedly limited knowledge of this area) novel. The key technical result is Theorem 1, which establishes a reduction in which one only needs to consider one fixed data distribution as an input into the policy, as opposed to stochastic observations generated from different distributions over time. This is perhaps not too surprising, but is non-trivial nonetheless. From here it's not that hard to get Theorem 2 using standard properties of Wasserstein/Kolmogorov distances. The paper then shows that for the newsvendor problem the rate obtained in Theorem 2 is tight, which is nice. The non-Lipschitz problem section is more interesting and surprising: under Kolmogorov uncertainty, SAA is still near-optimal for the pricing problem, however under Wasserstein uncertainty, SAA fails. Interesting, SAA can still be robustified to work under Wasserstein uncertainty. For the ski-rental problem, for both kinds of uncertainty balls SAA is highly suboptimal. Overall this is a fairly well-rounded analysis of the three motivating problems, which some general results (Thm 1/2) which are not too surprising but valuable nevertheless.

---

> ### Author Response · Authors · 2022-08-02
> **Response to Reviewer xMuo**
>
> We thank the reviewer for the assessment, the questions, and for the time spent reviewing our paper.
>
> *A natural question that arises is whether it is possible to somehow fix SAA to get the optimal rates for the ski-rental problem. I don't think this is necessary though.*
>
> **A:** If we understand the referee's question correctly, we clarify here that it is actually possible to ``fix'' SAA for the ski-rental problem, and this was shown in the submitted manuscript (although perhaps buried).
>
> For Wasserstein heterogeneity, we  design and analyze a $\delta$-SAA policy which inflates the action selected by SAA (i.e., $\delta > 0$) and that achieves the optimal dependence on $\epsilon$ and $M$. In the original manuscript, we only mentioned this in the appendix but in the revision, we also mention this in the main text (ln. 386) - the exact correction appears in Appendix 5.2.
>
> For Kolmogorov heterogeneity, our lower bound on SAA in Theorem 5, together with the result of Diakonikolas et al. (2021) allows us to conclude that a plain SAA can be highly suboptimal and a correction (such as the capping in Diakonikolas et al., 2021) is necessary.
>
> *The authors do not discuss limitations and potential negative societal impact explicitly. I don't see what they could say about negative societal impact, but perhaps they could add some discussion about limitations.*
>
> **A:** Thank you for your comment. In the revised version of our work, we extend the discussion in the conclusion and present several directions that, we think, could provide a better understanding of the performance of data-driven policies in heterogeneous environments (see ln. 423-433). We also discuss in section 5.1 some limitations of the inflated SAA policy in the pricing problem (knowledge of $\epsilon$, worse regret under Kolmogorov than SAA, see ln. 370-374).

---

> > ### Comment · Reviewer_xMuo · 2022-08-07
> > **Thank you for the response**
> >
> > Thank you for the response; the new Section 6 is indeed very helpful.

---

### Official Review · Reviewer_kSQ2 · 2022-07-21

**Rating:** 4
**Confidence:** 4
**Soundness:** 3 good
**Presentation:** 3 good
**Contribution:** 2 fair

**Summary:**

The paper proposes a new framework to quantify the deviation from historical (training) samples to future (test) samples. It considers three problems as special cases of the framework: newsvendor, pricing, and ski-rental. It examines sample-average-approximation (SAA) policy and derives upper and lower regret bounds. It also studies an inflation policy for the pricing problem where the SAA policy is suboptimal. The discussions are separated into two cases based on whether the underlying cost function is Lipschitz or not.

**Questions:**

See above.

**Ethics Review Area:**

["I don’t know"]

**Strengths And Weaknesses:**

The paper is well-written, and it proposes a new formulation for measuring the difference between historical and future samples.

My main concern is the technical contribution of the work, specifically, in the following aspects:
-  New policies: the new measure doesn't seem to elicit new/insightful policies other than SAA. The paper looks more like an analysis of SAA under a heterogenous environment where the test data differs from the training data. For the pricing problem, it proposes a $\delta$-SAA policy but other than that, all the upper bound analyses in this paper are based on the SAA policy. For the ski-rental problem, it mentioned: "... we derive a deviation policy which inflates the SAA action ...", but I didn't see the discussion of this inflated SAA policy (correct me if I missed anything here).
- Generalization to sequential or multi-variate settings: All three examples in the paper focus on a static setting with a single-dimensional random variable. Thus the technical results look straightforward to me. I wonder whether the results can be generalized to a sequential setting, or a setting with features/covariates.
- Robust optimization: In Theorem 1, the paper establishes distributionally robust optimization (DRO) as an asymptotic upper bound of the regret in this paper. For the finite-sample analysis, what if one adopts the DRO policy? Will that be sub-optimal or computationally intractable?

---

> ### Author Response · Authors · 2022-08-02
> **Response to Reviewer kSQ2**
>
> We are grateful for the referee's comments and questions, as well as the time invested in reviewing our paper.
>
> * *New policies: the new measure doesn't seem to elicit new/insightful policies other than SAA. The paper looks more like an analysis of SAA under a heterogeneous environment where the test data differs from the training data. For the pricing problem, it proposes a -SAA policy but other than that, all the upper bound analyses in this paper are based on the SAA policy. For the ski-rental problem, it mentioned: "... we derive a deviation policy which inflates the SAA action ...", but I didn't see the discussion of this inflated SAA policy (correct me if I missed anything here)*.
>
> **A:** Thank you for your comment. Our work indeed analyzes SAA under several notions of heterogeneity. Our interest for SAA stems from its wide use in practice but also from its strong performance in the i.i.d. regime.
>
> We would like to clarify, however, that we do not only analyze the SAA policy, but also analyze other policies. At a high level, we aim at deriving near-optimal policies, in the sense that these achieve a limiting minimax regret with optimal dependence in $\epsilon$ and $M$ as $\epsilon \downarrow 0$. It happens that SAA performs very well (in terms of achieving the correct dependence on $\epsilon$ and $M$) in some combinations of problem classes and heterogeneity types. However, when SAA fails, we analyze alternative policies. This happens for various combinations (e.g. in pricing and ski-rental under the Wasserstein distance). In such cases, we establish that  it is enough to consider small deviations to SAA (either deflate the decision (as in pricing) or inflate the decision (as in ski-rental)). Given that these problems have a poor structure (the objective function is not even continuous), we believe it is quite interesting that these small deviations are enough to achieve the best dependence in terms of $\epsilon$ and $M$.  The direction of deviation (down for pricing, up for ski rental) is also insightful.
>
> Concerning the inflated policy which achieves the optimal asymptotic worst-case regret for the Wasserstein ski-rental problem, we would like to clarify that it was included fully in the appendix of the original manuscript. In the revised version, we make more explicit in the main text  that this policy is a $\delta$-SAA policy with the correct choice of $\delta>0$ (see ln. 386). The exact value of $\delta$ is provided in appendix 5.2.
>
> * *Generalization to sequential or multi-variate settings: All three examples in the paper focus on a static setting with a single-dimensional random variable. Thus the technical results look straightforward to me. I wonder whether the results can be generalized to a sequential setting, or a setting with features/covariates.*
>
> **A:** While we agree with the referee that extending the results to settings with covariates is extremely interesting (we now comment on it explicitly in the concluding remarks), we will try to better highlight why we believe our technical contributions are important.
>
> First, we would like to note that the general reduction in Section 3 is not restricted to one-dimensional problems. Second, we would like to note that the one-dimensional problems studied are prototypical and are typically studied in isolation (given that they admit different structure). The present paper allows us to put them under the same umbrella and highlight significant differences across these problems.
>
> Finally, even when it comes to one-dimensional problems, to the best of our understanding, many of our results do not follow straight-forwardly. As mentioned earlier, for pricing and ski rental it is necessary to analyze policies beyond SAA in order to get good performance, and to our knowledge, this requires new techniques. We have tried to better emphasize this in the revision, and importantly, added a detailed  "Remark 2" on ln. 342-350  which explains why standard approaches fail.
>
> *Robust optimization: In Theorem 1, the paper establishes distributionally robust optimization (DRO) as an asymptotic upper bound of the regret in this paper. For the finite-sample analysis, what if one adopts the DRO policy? Will that be sub-optimal or computationally intractable?*
>
> **A:** We thank the reviewer for this question. Indeed, the connection to the DRO problem is interesting and suggests a systematic way to develop good policies in heterogeneous environments. While these policies should achieve an asymptotic optimal dependence in the problem parameters, it  is not clear if and how  these can be computed  efficiently and if one can derive some structure about these policies. In the revised version of our work, we emphasize more the interesting connection to DRO and state as an open question the analysis of the implied policies (see ln. 419-422).

---

> > ### Comment · Reviewer_kSQ2 · 2022-08-08
> > **Post-rebuttal**
> >
> > Dear authors,
> >
> > Thanks for the clarification. The revision does look much better.
> >
> > Given the other scores, I wouldn't be in objection to this paper going through. Yet I am personally still not quite convinced about the technical challenge of the analysis, say, after reading Line 342-350. So I would like to keep my score the same.

---

### Meta-Review · Area_Chair_wCQt · 2022-08-20

**Recommendation:** Accept
**Confidence:** Certain

**Metareview:**

This paper studies an interesting setting in which the future training data and the past data are generated from different distributions, but the two distributions are close in terms of Kolmogorov or Wasserstein distances.  The paper gives a nice set of algorithmic and impossibility results.  Initially, some reviewers had questions about the novelty of the framework and technical approaches; the authors have addressed most of these questions properly.

**Award:**

No

---

### Decision · Program_Chairs · 2022-09-14

Accept